# Intracerebral Administration of a Ligand-ASO Conjugate Selectively Reduces α-Synuclein Accumulation in Monoamine Neurons of Double Mutant Human A30P*A53T*α-Synuclein Transgenic Mice

**DOI:** 10.3390/ijms22062939

**Published:** 2021-03-13

**Authors:** Rubén Pavia-Collado, Valentín Cóppola-Segovia, Lluís Miquel-Rio, Diana Alarcón-Aris, Raquel Rodríguez-Aller, María Torres-López, Verónica Paz, Esther Ruiz-Bronchal, Leticia Campa, Francesc Artigas, Andrés Montefeltro, Raquel Revilla, Analia Bortolozzi

**Affiliations:** 1Institut d’Investigacions Biomèdiques de Barcelona (IIBB), Spanish National Research Council (CSIC), 08036 Barcelona, Spain; ruben.pavia@iibb.csic.es (R.P.-C.); lluis.miquel@iibb.csic.es (L.M.-R.); d.alarcon@keval.es (D.A.-A.); maria.torres@iibb.csic.es (M.T.-L.); veronica.paz@iibb.csic.es (V.P.); esther.ruiz@iibb.csic.es (E.R.-B.); lcmnqi@iibb.csic.es (L.C.); fapnqi@iibb.csic.es (F.A.); 2Institut d’Investigacions Biomèdiques August Pi i Sunyer (IDIBAPS), 08036 Barcelona, Spain; 3Centro de Investigación Biomédica en Red de Salud Mental (CIBERSAM), ISCIII, 28029 Madrid, Spain; 4Laboratory of Neurobiology and Redox Pathology, Department of Basic Pathology, Federal University of Paraná (UFPR), Curitiba 81531-980, Brazil; valen.coppola@gmail.com; 5CHU de Quebec Research Center, Axe Neurosciences. Department of Molecular Medicine, Faculty of Medicine, Université Laval, Quebec City, QC G1V 4G2, Canada; raquel.rodriguez-aller.1@ulaval.ca; 6CERVO Brain Research Centre, Quebec City, QC G1J 2G3, Canada; amontefeltro@lingeamc.com (A.M.); rrevilla@cubiqfoods.com (R.R.); 7n-Life Therapeutics, S.L., 18100 Granada, Spain

**Keywords:** α-synuclein, antisense oligonucleotide, dopamine neurotransmission, double mutant A30P*A53T*, motor deficits, Parkinson’s disease, transgenic mouse model

## Abstract

α-Synuclein (α-Syn) protein is involved in the pathogenesis of Parkinson’s disease (PD). Point mutations and multiplications of the α-Syn, which encodes the *SNCA* gene, are correlated with early-onset PD, therefore the reduction in a-Syn synthesis could be a potential therapy for PD if delivered to the key affected neurons. Several experimental strategies for PD have been developed in recent years using oligonucleotide therapeutics. However, some of them have failed or even caused neuronal toxicity. One limiting step in the success of oligonucleotide-based therapeutics is their delivery to the brain compartment, and once there, to selected neuronal populations. Previously, we developed an indatraline-conjugated antisense oligonucleotide (IND-1233-ASO), that selectively reduces α-Syn synthesis in midbrain monoamine neurons of mice, and nonhuman primates. Here, we extended these observations using a transgenic male mouse strain carrying both A30P and A53T mutant human α-Syn (A30P*A53T*α-Syn). We found that A30P*A53T*α-Syn mice at 4–5 months of age showed 3.5-fold increases in human α-Syn expression in dopamine (DA) and norepinephrine (NE) neurons of the substantia nigra pars compacta (SNc) and locus coeruleus (LC), respectively, compared with mouse α-Syn levels. In parallel, transgenic mice exhibited altered nigrostriatal DA neurotransmission, motor alterations, and an anxiety-like phenotype. Intracerebroventricular IND-1233-ASO administration (100 µg/day, 28 days) prevented the α-Syn synthesis and accumulation in the SNc and LC, and recovered DA neurotransmission, although it did not reverse the behavioral phenotype. Therefore, the present therapeutic strategy based on a conjugated ASO could be used for the selective inhibition of α-Syn expression in PD-vulnerable monoamine neurons, showing the benefit of the optimization of ASO molecules as a disease modifying therapy for PD and related α-synucleinopathies.

## 1. Introduction

In the elderly population, Parkinson’s disease (PD) is the second most common neurodegenerative disease after Alzheimer’s disease [1,2]. The illness is characterized by a variety of motor dysfunctions including difficulties in initiating movements, bradykinesia, rigidity, and resting tremor. These signs result from a reduction in striatal dopamine (DA) neurotransmission, which accompanies progressive degeneration of DA neurons in the substantia nigra pars compacta (SNc) [3,4,5,6]. However, PD is also characterized by a premotor phase associated with the development of neuropsychiatric symptoms, cognitive deficits, among others, involving dysfunctions of non-DA neurons, e.g., serotonin (5-HT) and norepinephrine (NE) neurons, which precedes the development of motor symptoms [7,8]. Although the causes of PD are still poorly understood, genetic studies have identified two highly penetrant mutations, A30P and A53T, in α-synuclein (α-Syn), which encodes *SNCA* gene associated with autosomal dominant inheritance of the PD [9,10]. α-Syn is a highly conserved protein made up of 140 amino acid residues that is predominantly expressed in neurons, and is abundantly localized in presynaptic terminals, which plays a significant role in the regulation of neurotransmitter release, synaptic function, and neuroplasticity [11,12,13].

Accumulation of α-Syn in intracytoplasmic inclusions called Lewy bodies is a neuropathological hallmark of PD. Therefore, one approach to modelling the disease is to modify the mouse genome in order to over-express wild-type human α-Syn (h-α-Syn), as well as the PD-linked α-Syn mutants A53T, A30P, and E46K, or even combinations of them [14,15,16,17,18,19]. Indeed, Richfield et al. [20] introduced a transgenic mouse model carrying a double mutant h-α-Syn gene with both A30P and A53T point mutations under the tyrosine hydroxylase (TH) promoter (A30P*A53T*α-Syn). Aged A30P*A53T*α-Syn mice (13–23 months) successfully recapitulated many important features of α-synucleinopathy, including increased α-Syn expression, which adversely induced an age-dependent DA neurodegeneration and a decrease in the DA concentration in the striatal tissue, leading to deficits in motor coordination. Moreover, A30P*A53T*α-Syn mice showed swollen, dystrophic TH^+^ neurites in the SNc, as well as accumulation of oligomeric α-Syn forms at 12 months, supporting the proposal that A30P*A53T*α-Syn mice may be used as a model to test new PD-modifying therapies that reduce α-Syn expression/accumulation [21,22].

In the last decade, remarkable advances in the development of oligonucleotide therapies aimed at inhibiting α-Syn synthesis have been made [23]. Gene silencing mechanisms targeting α-Syn mRNA may reduce the intracellular protein content and stop/slow the progression of the illness. Preclinical studies in rodents and nonhuman primates have successfully shown that α-Syn can be downregulated in PD-affected brain areas after direct application of oligonucleotide therapeutics including antisense oligonucleotides (ASO), small interfering RNAs (siRNA), and microRNAs (miRNA) [24,25,26,27,28,29,30,31]. Irrespective of these potential hitches, a major limitation in the development of oligonucleotide-based therapeutics is their delivery to the brain compartment, and once there, to selected neuronal populations or cell types. In an effort to solve this problem, we successfully developed a strategy to supply in vivo oligonucleotides selectively to brainstem monoamine neurons (DA, 5-HT, and NE). This was achieved by conjugating oligonucleotides with inhibitors of monoamine transporters (MAT) showing nM affinity for MAT, such as sertraline, reboxetine, and indatraline (IND), which are exclusively expressed in monoamine neurons at high densities [32,33,34,35,36]. MAT inhibitors allow the selective accumulation of oligonucleotides in monoamine neurons after internalization in deep Rab-7-associated vesicles [33,35].

Recently, we reported that intracerebroventricular or intranasal administration of an indatraline-conjugated 1233-ASO (IND-1233-ASO) can effectively reduce α-Syn protein accumulation in the brainstem monoamine neurons of wild-type mice and nonhuman primates, without causing neurotoxicity [34,36]. In the present study, we extended these observations and assessed whether an IND-1233-ASO sequence designed in such a way that the target mRNA sequence displays homology with the murine, rhesus macaque, and human α-Syn is able to downregulate h-α-Syn expression in DA and NE brain areas of transgenic A30P*A53T*α-Syn mice. Furthermore, since the A30P and A53T familial point mutations in the *SNCA* gene are a risk factor for early-onset PD [9,10,37], we also examined the anxiety-depressive phenotype and cognitive abnormalities, as well as DA function in middle-aged mice (5 months), as these features have not been assessed in previous studies using this transgenic mouse model.

## 2. Results

### 2.1. α-Syn Expression Profile in Brain Areas of A30P*A53T*α-Syn Transgenic Mice

We first examined the expression of h-α-Syn and murine α-Syn (m-α-Syn) mRNAs in several cortical and subcortical brain areas of non-transgenic (non-Tg) and transgenic A30P*A53T*α-Syn mice (Figure 1a and Appendix A). Using in situ hybridization, we found that h-α-Syn mRNA levels were more than 3-fold higher than m-α-Syn mRNA levels selectively in the brainstem nuclei containing DA and NE cell bodies in brain areas such as the SNc, ventral tegmental area—VTA and locus coeruleus—LC (SNc/VTA: 316.9% ± 18.3%; LC: 377.5% ± 22.5%, respectively, versus m-α-Syn mRNA levels) (Figure 1b). The Student’s t-test indicated values of *t* = 10.52, *p* < 0.0001 for SNc/VTA and *t* = 11.91, *p* < 0.0001 for LC, respectively. Moreover, m-α-Syn mRNA expression was unchanged, and comparable values were detected in cortical and subcortical brain areas of non-Tg and transgenic A30P*A53T*α-Syn mice (Appendix A). In parallel, A30P*A53T*α-Syn mice showed also significant reductions in murine γ-synuclein (γ-Syn) mRNA expression in the SNc/VTA and LC compared with non-Tg mice (*t* = 2.838, *p* = 0.0296; *t* = 3.005, *p* = 0.0239, Student’s *t*-test, respectively) (Figure 1c,d). The increases in h-α-Syn mRNA levels in the SNc/VTA and LC were confirmed by immunohistochemistry assessment of h-α-Syn protein levels (Figure 1e).

However, unlike the aged A30P*A53T*α-Syn mice (13–23 months), the middle-aged mice (5 months) did not show a loss of tyrosine hydroxylase (TH)^+^ cells in the SNc, VTA and LC, nor were there any changes in TH density in the caudate putamen (CPu) compared with non-Tg mice (Figure 2a,b). Likewise, no alterations were found using other DA and NE neuronal markers, including the DA transporter (DAT) and NE transporter (NET) proteins, respectively. Histochemical analysis showed that the DAT density levels in SNc/VTA and CPu result in similar levels of immunoreactivity for both phenotypes (Figure 2c,d). In addition, the NET density was comparable in the LC and medial prefrontal cortex (mPFC) of A30P*A53T*α-Syn and non-Tg mice (Figure 2e,f). Altogether, these results indicate that A30P*A53T*α-Syn overexpression, and concomitant reduction in γ-Syn levels in the SNc/VTA and LC occurred in the absence of monoamine neurodegeneration, at least in middle-aged mice.

### 2.2. A30P*A53T*α-Syn Transgenic Mice Show Motor Deficits and an Anxiety-Like Phenotype

To determine the functional consequences of A30P*A53T*α-Syn overexpression in SNc/VTA and LC, we performed a behavioral study assessing motor, emotional and cognitive components. Compared with non-Tg mice, A30P*A53T*α-Syn mice showed a reduced spontaneous locomotor activity as assessed in the open field test (Figure 3a,b). Differences were found between A30P*A53T*α-Syn and non-Tg mice in terms of total distance traveled, fast movements, mean speed, resting time, and vertical count. The total distance traveled and frequency of fast movements, but not slow movements, were significantly lower in A30P*A53T*α-Syn than non-Tg mice (*t* = 3.199, *p* = 0.0035; *t* = 2.675, *p* = 0.0125, Student’s *t*-test, respectively). Likewise, A30P*A53T*α-Syn mice exhibited a reduced mean speed (*t* = 3.105, *p* = 0.0044, Student’s *t*-test), and a longer resting time (*t* = 2.785, *p* = 0.0097, Student’s *t*-test) compared with non-Tg mice. Vertical counts were also significantly lower in A30P*A53T*α-Syn mice than in non-Tg mice both in the open field test (Figure 3a, *t* = 2.91, *p* = 0.0074, Student’s *t*-test) and cylinder test (Figure 3c, *t* = 2.397, *p* = 0.0247, Student’s *t*-test). However, A30P*A53T*α-Syn mice showed no difference compared to non-Tg mice in motor asymmetry in the cylinder test as expected (Figure 3c).

Mice were also tested using a dark-light box paradigm, a behavioral task assessing the anxiety-like phenotype (Figure 3d). The A30P*A53T*α-Syn mice showed a significant reduction in time spent in the light box (*t* = 2.162, *p* = 0.0404, Student’s *t*-test), and there were marginal effects on latency and the number of entries in the light box, suggesting an anxious phenotype compared with non-Tg mice. Nevertheless, no differences between both phenotypes were detected in the tail suspension test or in the novel object recognition, behavioral tasks assessing vulnerability/resilience to stress and short-term memory, respectively (Appendix A).

### 2.3. A30P*A53T*α-Syn Overexpression in TH^+^ Neurons Impairs DA Neurotransmission in the Nigrostriatal Pathway

The impact of the mutant α-Syn transgene overexpression on the nigrostriatal DA function was examined in 5-month-old A30P*A53T*α-Syn and non-Tg mice using in vivo microdialysis procedures. A30P*A53T*α-Syn transgenic mice showed no differences in baseline extracellular concentrations of DA and its 3,4-dihidroxyphenylacetic acid (DOPAC) metabolite in CPu compared with non-Tg mice (Table 1). However, infusion of the depolarizing agent veratridine (50 μM) by reverse dialysis increased extracellular DA in CPu to a lesser extend in A30P*A53T*α-Syn than in non-Tg mice (~2-fold versus ~6-fold, respectively) (Figure 4a). Two-way ANOVA followed by Tukey’s post hoc test analysis showed a marginal effect of group F_(1,7)_ = 3.993, *p* = 0.0858, and effect of time F_(15,105)_ = 19.85, *p* < 0.0001 and a group-by-time interaction F_(15,105)_ = 3.304, *p* = 0.0002.

Moreover, both α-Syn and γ-Syn proteins modulate membrane distribution and the reuptake function of DAT at DA terminals [19,34]. Local application of amphetamine (DA releaser and DAT inhibitor, 1–10 μM) dose-dependently elevated the dialysate DA concentration in the CPu, being significantly higher in A30P*A53T*α-Syn transgenic mice compared with non-Tg mice (Figure 4b). Two-way ANOVA showed effects of group F_(1,126)_ = 7.828, *p* = 0.0060 and time F_(17,126)_ = 6.235, *p* < 0.0001, and a marginal effect of group-by-time interaction F_(17,126)_ = 1.560, *p* = 0.0849. However, no significant differences were observed in the extracellular DA concentration in the CPu between both phenotypes after local application of DAT inhibitor nomifensine (1–10 μM) (Figure 4c), suggesting that the greater DA elevation induced by amphetamine in A30P*A53T*α-Syn transgenic mice is not due to a differential expression/function of DAT. Two-way ANOVA showed an effect of time F_(17,162)_ = 7.474, *p* < 0.0001, but not of group as well as a group-by-time interaction.

To obtain more information on the mechanisms controlling DA neurotransmission in the nigrostriatal pathways of A30P*A53T*α-Syn mice, we also examined the effect of the DA D2 receptor agonist quinpirole on DA release. Local quinpirole infusion (10 μM) by reverse dialysis comparably reduced the extracellular DA concentration in the CPu of non-Tg and A30P*A53T*α-Syn mice (Figure 4d). Two-way ANOVA showed an effect of time F_(17,144)_ = 11.20, *p* < 0.0001, but not of group as well as a group-by-time interaction. No significant differences in extracellular DOPAC levels were observed between both phenotypes with the different pharmacological approaches used (data not shown) [19].

### 2.4. ASO Therapy Reduces the Accumulation of Human α-Syn and Normalizes DA Neurotransmission Deficits but Not the Behavioral Phenotype in A30P*A53T*α-Syn Transgenic Mice

Recently, we reported that IND-1233-ASO selectively reduced murine α-Syn expression in monoaminergic neurons of wild-type mice and nonhuman primates [34,36]. Likewise, the IND-1337-ASO sequence, designed specifically to target h-α-Syn in a PD-like mouse model overexpressing wild-type h-α-Syn induced by an adeno-associated viral vector, also prevented h-α-Syn accumulation in interconnected DA brain regions [36]. This specificity is due to the potent in vitro affinity and in vivo occupancy of IND for monoamine transporters [38], which allows the accumulation of the conjugated oligonucleotide in this neuronal population after intracerebroventricular or intranasal administration. Here, we extended these previous studies, and confirmed the intracellular accumulation of IND-conjugated oligonucleotides, specifically in TH^+^ neurons of the SNc/VTA and LC of A30P*A53T*α-Syn mice after a single intracerebroventricular application of IND-1233-ASO (100 μg/mouse) (Figure 5a). In addition, IND-1233-ASO administration (100 μg/day for 28 days) significantly reduced mutant α-Syn transgene expression in the SNc/VTA and LC compared with A30P*A53T*α-Syn mice treated with a vehicle (Figure 5b). The reduction in h-α-Syn mRNA was ~54% and ~32% in the SNc/VTA and LC, respectively, compared to the level in A30P*A53T*α-Syn mice receiving a vehicle (Figure 5c). Two-way ANOVA showed an effect of treatment F_(1,12)_ = 42.84, *p* < 0.0001, α-Syn phenotype F_(1,12)_ = 99.71, *p* < 0.0001 and treatment-by- α-Syn phenotype interaction F_(1,12_) = 42.84, *p* < 0.0001 for SNc/VTA, as well as effects of treatment F_(1,12)_ = 10.06, *p* = 0.0080 and α-Syn phenotype F_(1,12)_ = 157.3, *p* < 0.0001 and a treatment-by-α-Syn phenotype interaction F_(1,12)_ = 17.47, *p* = 0.0013 for LC. Interestingly, IND-1233-ASO treatment did not alter m-α-Syn mRNA expression (Figure 5c). The decreased h-α-Syn mRNA level in the SNc/VTA and LC was accompanied by a significant decrease in the h-α-Syn protein level, as assessed by immunohistochemistry (SNc/VTA: *t* = 3.047, *p* = 0.00226, LC: *t* = 5.812, *p* = 0.0011, Student’s *t*-test) (Figure 5d,e). Likewise, a lower number of h-α-Syn^+^ cells was found in the SNc/VTA, but not in the LC, of A30P*A53T*α-Syn mice treated with IND-1233-ASO vs. those treated with the vehicle (*t* = 7.163, *p* = 0.0004, Student’s *t*-test) (Figure 5e). Furthermore, h-α-Syn protein accumulation also decreased in brain areas with dense DA and NE innervation, such as the CPu (t = 2.809, *p* = 0.0376, Student’s *t*-test) and mPFC (*t* = 2.591, *p* = 0.0411, Student’s *t*-test) (Figure 5d,e).

In parallel, the IND-1233-ASO-induced decrease in h-α-Syn expression normalized γ-Syn mRNA expression in the SNc/VTA (marginal effect) and LC of A30P*A53T*α-Syn mice compared to those mice treated with a vehicle (Figure 5f,g). One-way ANOVA showed an effect of group in SNc/VTA (F_(2,6)_ = 14.01, *p* = 0.0055) and LC (F_(2,7)_ = 8.575, *p* = 0.0131), respectively. Remarkably, IND-1233-ASO treatment did not induce any changes in TH, DAT, and NET protein levels in the SNc/VTA, or LC or in DA/NE projection brain areas, which supports the specificity and safety of the IND-1233-ASO sequence (Appendix A). Similarly, IND-1233-ASO was well tolerated during the 4 weeks of treatment, and no changes were found in daily observed behavioral variables, such as water and food consumptions, as well as body weight in A30P*A53T*α-Syn transgenic mice compared to vehicle-treated A30P*A53T* α-Syn mice (data not shown), as previously reported [34,36].

Next, we found that A30P*A53T*α-Syn transgenic mice treated with IND-1233-ASO showed normalization of DA neurotransmission in the nigrostriatal pathway, reaching extracellular DA levels similar to those detected in non-Tg mice (Figure 4a,b and Figure 6a,b). Local veratridine administration (50 μM) significantly increased striatal DA release in A30P*A53T*α-Syn mice treated with IND-1233-ASO compared with transgenic mice receiving a vehicle (Figure 4a and Figure 6b). Two-way ANOVA showed an effect of time F_(15,80)_ = 10.23, *p* < 0.0001 and a group-by-time interaction F_(15,80)_ = 2.021, *p* = 0.0235, but no effect of group. Likewise, local amphetamine infusion (1–10 μM) increased extracellular DA levels to a lesser extent in the CPu of A30P*A53T*α-Syn mice treated with IND-1233-ASO compared with those treated with vehicle (Figure 4a and Figure 6b). Two-way ANOVA showed an effect of group F_(1,72)_ = 7.359, *p* = 0.0083 and time F_(17,72)_ = 4.231, *p <* 0.0001, but no group-by-time interaction. However, despite the changes in DA neurotransmission, treatment with IND-1233-ASO for 28 days was insufficient to significantly reverse the behavioral impairments observed in A30P*A53T*α-Syn mice, and the mice exhibited motor alterations and anxiety-like behavior similar to those that received the vehicle (Figure 6c,d).

## 3. Discussion

The present study confirms and extends the results of previous studies on the benefit of a sustained therapy based on an IND-1233-ASO conjugate to reduce h-α-Syn expression selectively in early affected DA and NE brain areas –SNc/VTA and LC– in PD, as assessed in a transgenic mouse carrying both A30P and A53T mutant h-α-Syn. Remarkably, IND-1233-ASO also decreased h-α-Syn protein accumulation in DA/NE projection cortical and subcortical brain areas such as mPFC and CPu in the same mice. Likewise, no signs of toxicity were found in DA and NE neurons as assessed with classical DA and NE neuronal markers, in line with previous reports using wild-type mice and elderly nonhuman primates [34,36]. Human α-Syn knockdown in monoaminergic nuclei normalized the nigrostriatal DA neurotransmission in the A30P*A53T*α-Syn transgenic mice, although the current IND-1233-ASO therapy could not significantly reverse the behavioral phenotype, perhaps due to an insufficient time/dose and/or the involvement of downstream changes in relevant brain circuits not recovered by the treatment. Therefore, the current strategy of using ligand-oligonucleotide conjugates appears to have the potential for future optimization of new ligand-conjugated oligonucleotide-based therapies to treat PD and related α-synucleinopathies.

Selective reduction in h-α-Syn expression was found in the DA and NE brain regions of transgenic mice by exploiting our previous strategy, in which different oligonucleotide sequences were covalently bound to MAT inhibitors for the selective delivery of them to monoaminergic cells in vivo [32,33,34,35,36]. The potential use of oligonucleotide therapeutics, such as ASO, for brain illness has aroused a great deal of interest; however, delivery to the brain compartment remains a key obstacle. The issue is not simply getting oligonucleotides to the brain but getting them to specific brain cells and to intracellular sites where they actually function. Remarkably, the current IND-1233-ASO conjugate was capable to accumulate the 1233-ASO sequence in DA and NE neurons of A30P*A53T*α-Syn mice after intracerebroventricular administration, taking advantage of by the dense network of axons emerging from monoamine cell bodies containing the greatest DAT and NET densities, for which IND has a high in vitro affinity [38,39]. Both DAT and NET are subject to modulation by the α-Syn protein and, to a lesser extent, by γ-Syn in cellular trafficking models [40]. Although, the A30P*A53T*α-Syn transgenic mice showed upregulated h-α-Syn levels and concomitantly downregulated endogenous m-γ-Syn levels, with no change to m-α-Syn levels, the total density of DAT/NET in the SNc/VTA, LC, and brain projection areas was comparable to that of non-Tg mice. Likewise, DAT expression/function was similar in both phenotypes, since the DAT inhibitor nomifensine induces the same effect on DA uptake, which suggests that DAT is functionally active in transgenic mice and is capable of facilitating IND-1233-ASO accumulation in DA neurons. Indeed, our previous studies showed that a reduction in MAT expression/function by 50% still allowed ASOs/siRNAs conjugated with MAT inhibitors to be internalized efficiently into monoaminergic neurons through a Rab7-dependent endosomal mechanism [33,34,35].

In the present study, intracerebroventricular IND-1223-ASO treatment (100 µg/mouse/day, 28 days) prevented the accumulation of h-α-Syn mRNA and protein in the SNc/VTA and LC of transgenic mice, without affecting endogenous m-α-Syn mRNA level, unlike a previous study in which the IND-1233-ASO sequence reduced m-α-Syn expression in wild-type mice [34]. The 1233-ASO sequence was designed to efficiently target homologous α-Syn mRNAs, including those of mice, monkeys and humans, as described previously in models expressing a single α-Syn phenotype [34,36]. Therefore, it is conceivable that the 1233-ASO sequence interacts with a different binding potency against homologous α-Syn mRNAs expressed at different concentrations in the same biological system as was used here, where h-α-Syn mRNA has 3.5-fold greater expression than that of m-α-Syn mRNA. Moreover, the high efficiency of the silencing depends on the stability of the mRNA/DNA oligonucleotide hetero-duplex, perhaps higher for h-α-Syn vs. m-α-Syn mRNA, as previously reported [41].

In any case, the 30–40% decrease in h-α-Syn mRNA levels obtained after intracerebroventricular treatment did not induce DA and NE neurotoxicity nor striatal and cortical denervation as assessed by TH, DAT or NET immunohistochemistry. However, using short-hairpin RNA (shRNA) cloned in adeno-associated viral (AAV) vector to knockdown α-Syn (>80% of decrease), some reports indicated DA neurodegeneration and loss of striatal DA fibers in rats and nonhuman primates [42,43,44,45], suggesting the need to maintain a threshold for α-Syn knockdown since this protein is essential for neurotransmission homeostasis, among other biological functions. These data are important in the context of ongoing attempts to develop PD therapeutics targeting α-Syn expression. Although viral vector-mediated antisense therapies such as AAV, are useful for delivering antisense/shRNA sequences into cells; currently, they have certain issues involving (i) genome integration, (ii) the inability to be delivered repeatedly, and (iii) possible host rejection. Alternatives to viral strategies are currently being developed based on non-viral oligonucleotides (siRNA, ASO, miRNA). Non-viral approaches are biologically safer and much less immunogenic than viral vectors [46,47,48,49]. Conversely, for non-viral oligonucleotides to be effective, they must achieve delivery to affected regions or cells at appropriate concentrations. Furthermore, they must be stable requiring additional protection measures (end-blocking, base modification, etc.), and maintain efficacy over time for feasible treatment throughout the course of disease [49,50,51]. More recently, there is a surge of interest in the delivery of oligonucleotides to specific cell types in the brain using ligand-oligonucleotide conjugates as reported here. This cellular selectivity may be achieved by conjugating oligonucleotides to a ligand that interacts selectively with a receptor/transporter expressed in the cellular surface. Some ligand candidates that demonstrate high affinity and selectivity for brain receptors include anisamide or anandamide, ligands that bind to the sigma or cannabinoid receptors, respectively, providing effective functional delivery of ASO/siRNA to several neuronal types [52,53]. However, none of these studies allowed the delivery of oligonucleotides to specific neuronal populations. In contrast, we successfully developed a strategy to supply in vivo oligonucleotides selectively to brainstem monoamine neurons ([32,33,34,35,36] and present study). Therefore, although the conjugated ligand-oligonucleotide approach is still under development, it offers a promising pathway for oligonucleotide therapeutics for brain diseases.

Previous studies showed that the A30P*A53T*α-Syn transgenic mouse line on the same genetic background (C57BL/6J, α-Syn-39 line) used here exhibits significant age-dependent abnormalities in the locomotor activity from 7 months onwards [20,22]. Furthermore, the presence of the pathogenic A53T*α-Syn mutation in different lines of transgenic mice led to the development of severe motor impairment resulting in death at 8–12 months of age, while in younger animals altered motor activity and anxiety-like behaviors have also been reported [14,15,18,19]. These behavioral changes have been associated with a reduced tissue DA concentration and impaired synaptic plasticity seen within the basal ganglia prior to obvious neurodegeneration [19,54]. Our results also clearly showed that the A30P*A53T*α-Syn overexpression in the DA and NE brain areas leads to motor deficits and anxiety-like behaviors but not depressive-like phenotype nor cognitive dysfunction, in 5-month-old transgenic mice. In parallel, the A30P*A53T*α-Syn mice exhibited an altered DA neurotransmission in the nigrostriatal pathways (the effects on NE release were not assessed because of a lack of ability to perform adequate high-performance liquid chromatography-HPLC methods in our laboratory), without DA and NE neurodegeneration. These above observations emphasize the fact that the behavioral phenotype of DA deficits is not only a result of cell death and denervation, but that functionally impaired neurons contribute to the outcome and should be considered as a target for treatment. Thus, a potential attenuation of α-Syn expression/function could contribute to the reduction in deficits in DA neurotransmission, since α-Syn modulates neuronal DA activity [34]. In fact, IND-1233-ASO treatment (100 µg/mouse/day, 28 days) not only prevented h-α-Syn protein accumulation in SNc/VTA, LC, and anatomically connected brain areas but also recovered the DA neurotransmission, allowing it to reach the levels of non-Tg mice and normalize the expression of m-γ-Syn, which is reduced in A30P*A53T*α-Syn mice. However, this was not sufficient to improve the behavioral phenotype, suggesting a need to improve the pharmacokinetic and pharmacodynamic properties of conjugated oligonucleotide-based therapies, which are still poorly known, but which have far-reaching implications for dosing regimens and clinical efficacy.

A key observation in the present study is that the overexpression of A30P*A53T*α-Syn had opposing effects on DA release in the nigrostriatal terminals after pharmacological stimulation. The depolarizing agent veratridine induced a lower increase in extracellular DA in A30P*A53T*α-Syn mice than in non-Tg mice, whereas amphetamine (DA releaser and DAT inhibitor) produced robust increases in extracellular DA levels in both groups, which were greater in A30P*A53T*α-Syn mice. The effects of veratridine on DA release are in full agreement with previous findings which indicated that nigrostriatal terminals of α-Syn-deficient mice display standard tonic DA activity after electrical stimulation with single pulses; however, they exhibited increased DA release with paired stimuli that elevated intracellular Ca^2+^ levels and vice versa in mice overexpressing α-Syn [55]. Indeed, we also demonstrated that the overexpression of wild-type h-α-Syn in the SNc using an AAV vector or in transgenic mice leads to less veratridine-dependent DA release, although the basal DA levels did not change compared with control groups [34,36].

In contrast, striatal amphetamine application significantly increased DA release in A30P*A53T*α-Syn mice compared with non-Tg mice, unlike in another wild-type α-Syn overexpression mouse models [34,36]. Given that DAT expression/function was comparable in both groups of mice (e.g., similar effect of the DAT inhibitor nomifensine), the differences between veratridine and amphetamine may be due to differences in DA releasable pools in A30P*A53T*α-Syn mice. Hence, while veratridine releases the impulse-dependent DA vesicular pool, amphetamine also releases a cytoplasmic, impulse-independent DA pool. Overall, these data suggest that there is impaired DA storage in the synaptic vesicles of A30P*A53T*α-Syn transgenic mice. This possibility is consistent with the role of α-Syn in regulating synaptic vesicle release or uptake at the monoamine terminals. In vitro studies have shown that both wild-type α-Syn and the mutant A30P*α-Syn attenuated DAT function, as well as DAT trafficking away from the plasma membrane. Interestingly, the mutant A53T*α-Syn was unable to modulate DAT function, and subsequent studies showed that this protein only very weakly interacted with the transporter [40,56,57]. Early studies also reported that the A53T* mutant can modulate the vesicular monoamine transporter (VMAT2) [56,58]. In MESC2.10 human mesencephalic cells, the presence of the A53T* mutant decreased the expression of VMAT2, accompanied by decreased potassium-induced and increased amphetamine-induced DA release, respectively [59]. These observations suggest that the A53T*α-Syn, through its effects on VMAT2, impairs vesicular DA storage and release. Altogether, this evidence supports the different in vivo effects on extracellular DA of nomifensine (DAT inhibitor) and amphetamine (DA releaser and DAT inhibitor) in A30P*A53T*α-Syn mice. In line with this observation, amphetamine-induced locomotor activity was increased in the A30P*A53T*α-Syn mouse line at 6 months of age [22].

In conclusion, the cellular selectivity and efficacy obtained with IND-1233-ASO indicate that the ligand-oligonucleotide approach has great potential to reduce wild-type and mutant human α-Syn expression specifically in monoamine neurons, with a high translational value for the treatment of PD and other α-synucleinopathies. The reduction in α-Syn synthesis did not cause monoaminergic neurodegeneration and actually improved forebrain DA neurotransmission. This study shows the benefit of optimizing conjugated ASO molecules as a disease-modifying therapy for PD, something that might be attractive in conjunction with current immunotherapy trials targeting α-Syn protein, or those with anti-aggregation agents.

## 4. Materials and Methods

### 4.1. Animals

Adult male C57BL/6J transgenic mice carrying both A30P and A53T mutant human α-Syn under TH promotor (A30P*A53T*α-Syn-39Eric/J referred to as A30P*A53T*α-Syn, 4–5 months; The Jackson Laboratory, USA) were housed under standard laboratory conditions (12 h light/dark cycle; room temperature, 23 ± 2 °C; relative humidity 50 ± 15%) with food and water available ad libitum. In addition, non-transgenic littermates with the same genetic background were used as the control group (referred to as non-Tg). We used 105 mice for the whole study. Animal procedures were conducted in accordance with standard ethical guidelines (EU directive 2010/63 of 22 September 2010) and approved by the local ethical committee (University of Barcelona).

### 4.2. Conjugated Antisense Oligonucleotide

The synthesis and purification of indatraline-conjugated ASO molecule targeting α-Syn (IND-1233-ASO, GenBank accession AH008229.3, NCBI, Bethesda, MD, USA) were performed by nLife therapeutics S.L., as previously reported [32,33]. ASO consistent of an antisense GapMer of 18-mer single stranded DNA molecule with four 2′-*O*-methyl RNA bases at both ends to protect the internal DNA from nuclease degradation and improve the binding to the target mRNA. The sequence of IND-1233-ASO is 5′-cuccdCdTdCdCdAdCdTdGdTdCuucu-3′, where chemical modifications of the backbone including 2′-*O*-methyl are shown in lowercase letters, while “d” indicates the presence of 2′-deoxynucleotides (Appendix A). In brief, ASO synthesis was performed using ultra mild-protected phosphoramidites (Glen Research, Sterling, VA, USA) and an H-8 DNA/RNA automatic synthesizer (K&A Laborgeraete GbR, Schaafheim, Germany). Indatraline hydrochloride (triple blocker of monoamine transporters) was conjugated to 5′-carboxy-C10 modified oligonucleotide through an amide bond. This condensation was carried out under organic conditions (DIPEA/DMF, 24 h). Conjugated oligonucleotides were purified by high performance liquid chromatography (HPLC) using a RP-C18 column (4.6 × 150 mm, 5 µm) under a linear gradient condition of acetonitrile. The molecular weight of the oligonucleotide strands was confirmed by MALDI-TOF mass spectrometry (Ultra-flex, Bruker Daktronics, Billerica, MA, USA). The concentration of conjugated sequences was calculated based on the absorbance at a wavelength of 260 nm. In addition, to study the in vivo brain distribution and intracellular incorporation of IND-1233-ASO into DA and NE neurons, IND-1233-ASO molecules were additionally bound to fluorophore Alexa488 as previously reported [33,34]. Stock ASO solutions were prepared in RNAse-free water and stored at −20 °C until use.

### 4.3. Mouse Treatments

For intracerebroventricular administration, mice underwent pentobarbital anesthetization (40 mg/kg, intraperitoneal) and were placed in a stereotaxic frame. Buprenorphine (0.05 mg/kg, subcutaneous) was used as an analgesic. A micro-osmotic pump (Alzet Model 1004, Durect Corporation, Cupertino, CA, USA) was subcutaneously implanted, while the cannula (Brain Infusion Kit 3, Alzet) was implanted in the lateral ventricle (antero-posterior −0.34, medial-lateral −1.0 and dorsal-ventral −2.2 in mm) [34,60]. Micro-pumps were filled with IND-1233-ASO (37.87 mg/mL). Alzet model 1004 osmotic pumps deliver an average infusion of 0.11 µL/h (2.64 µL/day) for 28 days, providing an IND-1233-ASO dose of 100 µg/day (16.3 nmol/day). We also added a control group that received the vehicle (2.64 µL/day, artificial cerebrospinal fluid, in nM: NaCl, 125; KCl, 2.5; CaCl_2_, 1.26 and MgCl_2_, 1.18 with 5% glucose). The IND-1233-ASO sequence has been extensively characterized in previous studies, and the dose was chosen because it was safe without signs of neuronal and glial toxicity compared within IND-conjugated nonsense ASO sequence (IND-1227-ASO) [34,36].

To confirm the accumulation of IND-1233-ASO in DA and NE neurons, some mice (*n* = 3) received a single dose of IND-1233-ASO linked to Alexa488 (100 µg, 2.5 µL) in the lateral ventricle, and the mice were sacrificed 24 h later, as previously reported [33,34].

Mice were randomly assigned to treatment groups. After surgery, the animals were kept in individualization cages and the well-being of each mouse was monitored for the duration of the procedure. The body weight and general appearance of the mice were controlled according to Morton and Griffiths protocols [61].

### 4.4. In Situ Hybridization

Mice were killed by pentobarbital overdose and brains were rapidly removed, frozen on dry ice and stored at −80 °C. Coronal brain sections containing the SNc, VTA, LC, and projection brain areas including mPFC, hippocampus (HPC), and CPu (14 μm-thick) were obtained and processed, as described elsewhere [32,33]. Antisense oligonucleotides were complementary to bases, as follows: mouse α-Syn (m-α-Syn, 411-447 sequence, GenBank accession NM_001042451), human α-Syn (h-α-Syn, 2498-2548 sequence, NM_000345), and mouse γ-Syn (m-γ-Syn, 366-416 sequence, NM_011430 (IBA Nucleic Acids Synthesis). Oligonucleotides were individually labeled (2 pmol) at the 3′-end with ^33^P-dATP (>2500 Ci/mmol; DuPont-NEN) using terminal deoxynucleotidyl-transferase (TdT, Calbiochem). For hybridization, the radioactively labeled probes were diluted in a solution containing 50% formamide, 4× standard saline citrate, 1× Denhardt’s solution, 10% dextran sulfate, 1% sarkosyl, 20 mM phosphate buffer, pH 7.0, 250 μg/mL yeast tRNA, and 500 μg/mL salmon sperm DNA. The final concentrations of radioactive probes in the hybridization buffer were in the same range (~1.5 nM). Tissue sections were covered with hybridization solution containing the labeled probes, overlaid with para-film coverslips and, incubated overnight at 42 °C in humid boxes. Sections were washed four times (45 min each) in a buffer containing 0.6 M NaCl and 10 mM Tris-HCl (pH 7.5) at 60 °C. Hybridized sections were exposed to Biomax-MR film (Kodak, Sigma-Aldrich, Madrid, Spain) for 24–72 h with intensifying screens. For specificity control, adjacent sections were incubated with an excess (50×) of unlabeled probes. Films were analyzed and relative optical densities were evaluated in three adjacent sections in duplicate for each mouse and averaged to obtain individual values using a computer-assisted image analyzer (MCID, Mering, Germany). The MCID system was also used to acquire black and white images. Figures were prepared for publication using Adobe Photoshop software (Adobe Software, San José, CA, USA). Contrast and brightness of images were the only variables that were adjusted digitally.

### 4.5. Immunohistochemistry and Immunofluorescence

Mice were anaesthetized with pentobarbital and transcardially perfused with 4% paraformaldehyde (PFA) in sodium-phosphate buffer (pH 7.4). Brains were extracted, post-fixed 24 h at 4 °C in the same solution, and placed in gradient sucrose solution 10–30% for 3 days at 4 °C. After cryopreservation, serial 30 μm-thick sections were cut to obtain SNc, VTA, LC, mPFC and CPu. Brain sections were washed and incubated in a 1x PBS/Triton 0.2% solution containing normal serum from the secondary antibody host. Primary antibodies for h-α-Syn (anti-h-α-Syn clone Syn211, 1:1350; ref.: MA1-12874, Thermo Fisher Scientific, Waltham, MA, USA), DAT (anti-DAT, 1:2500 for SNc/VTA and 1:5000 for CPu; ref.: MAB369, Millipore, Burlington, MA, USA), and TH (anti-TH, 1:5000; ref.: AB112, Abcam, Cambridge, UK) were used. Briefly, primary antibodies were incubated overnight at 4 °C, followed by incubation with the corresponding biotinylated anti-mouse IgG1 (1:200, ref.: A-10519, Life Technologies, Carlsbad, CA, USA) for anti-h-α-Syn, biotinylated anti-rat IgG (1:200, ref.: BA-9401, Vector Laboratories, Burlingame, CA, USA) for anti-DAT, and biotinylated anti-rabbit IgG (1:200; ref.: BA-1000, Vector Laboratories) for anti-TH according to the manufacturer’s instructions. The color reaction was performed thorugh incubation with diaminobenzidine tetrahydrochloride (DAB) (ref.: D5905-50TAB, Sigma-Aldrich) solution. Sections were mounted and embedded in Entellan (Electron Microscopy Sciences, Hatfield, PA, USA).

In addition, primary antibodies included mouse anti-NET (1:1000, ref.: NET05-2 clone 2-3B2scD7, Mab Technologies, Stone Mountain, GA, USA). Slides were washed multiple times with PBS and incubated with fluorescently labeled secondary antibody A555 anti-mouse IgG (1:500, ref.: A-21427, Life Technologies) for 2 h at room temperature. Immunofluorescent slides were treated for 10 min with Hoechst dye (1:10,000) before the final PBS wash and cover-slipping with Fluoromont Aqueous Mounting Medium (Sigma-Aldrich, F4680).

The number of TH-positive cells and h-α-Syn-positive cells and their intracellular densities in the SNc/VTA and LC were assessed in sections corresponding to different antero-posterior levels −2.70 to −3.80 mm and −5.34 to −5.80 mm from the bregma, respectively using ImageJ software (v1.51s, NIH, Bethesda, MD, USA). All labeled cells with its nucleus within the counting frame were bilaterally counted in six consecutive SNc/VTA and three consecutive LC sections, and three different microscope fields were analyzed in each section. In addition, the intensity of h-α-Syn labelling in the CPu and mPFC, as well as the DAT density in SNc/VTA and CPu, and NET density in LC and mPFC were quantified in four consecutive sections corresponding to different antero-posterior levels (mm from bregma): 2.34 to 1.70 for mPFC, 1.70 to 0.02 for CPu, −2.70 to −3.80 for the SNc/VTA and −5.34 to −5.80 for LC, respectively, using ImageJ software (v1.51s).

### 4.6. Confocal Fluorescence Microscopy

Mice were sacrificed 24 h after intracerebroventricular Alexa-488 labelled IND-1233-ASO administration, and their brains were extracted and processed for immunofluorescence. Brain sections were rinsed with PBS/Triton 0.2%, incubated with normal serum from the secondary antibody host and treated with primary antibody rabbit anti-TH (1:1250; ref.: AB112, Abcam). Sections were then incubated overnight at 4 °C, rinsed and treated with secondary Alexa555-conjugated antibody (donkey anti-rabbit; 1:500; ref.: A-31572, Life Technologies) for 2 h. After subsequent washes, the sections were dehydrated, and mounted in the anti-fading agent Prolong Gold with DAPI (Life Technologies). DAPI, Alexa488 and Alexa555 images were acquired sequentially using a Leica TCS SP5 laser scanning confocal microscope (Leica Microsystems Heidelberg GmbH, Manheim, Germany) with excitation at 405, 488 nm and 553 nm laser line, respectively. Images were obtained at 400Hz in a 1024 × 1024 pixels format and were composed using ImageJ software (v1.51s).

### 4.7. In Vivo Microdialysis

Extracellular DA and DOPAC concentrations in the nigrostriatal pathway were measured by in vivo microdialysis as described elsewhere [34,36]. One concentric dialysis probe (Cuprophan membrane; 6000 Da molecular weight cut-off; 2 mm-long) was implanted in the mouse CPu (antero-posterior 0.5, medial-lateral −1.7 and dorsal-ventral −4.5, in mm) [60]. Experiments were performed 24–48 h after surgery in freely moving mice. In addition, the transgenic A30P*A53T*α-Syn mice treated with IND-1233-ASO or a vehicle (intracerebroventricular using osmotic mini-pumps) also had a guide cannula (CMA 7, ref.: CMAP000138, Harvard Apparatus, Holliston, MA, USA) implanted into the CPu. A dummy cannula was inserted into the guide cannula to avoid contamination, and this was fixed with screws. The day prior to the microdialysis experiments, the dummy cannula was removed and replaced with a 2 mm long microdialysis probe (CMA 7 microdialysis probe, ref.; CMAP000083, Harvard Apparatus). Microdialysis experiments were performed by an experimenter who was blinded to mouse treatments. DA and DOPAC levels in dialysate samples were analyzed using HPLC coupled with electrochemical detection (+0.7 V, Waters 2465, Milford, MA, USA), with 3-fmol detection limit. The mobile phase containing 0.15 M NaH_2_PO_4_.H_2_O, 0.9 mM PICB8, 0.5 mM EDTA (pH 2.8 adjusted with orthophosphoric acid), and 10% methanol was pumped at 1 mL/min (Waters 515 HPLC pump). DA and DOPAC were separated on a 2.6 μm particle size C18 column (7.5 × 0.46 cm, Kinetex, Phenomenex, Torrance, CA, USA) at 28 °C.

All reagents used were of analytical grade and were obtained from Merck (Darmstadt). DA hydrochloride, DOPAC acid, (–)-quinpirole hydrochloride, and nomifensine maleate were sourced from Sigma-Aldrich-RBI. D-amphetamine sulphate and veratridine were purchased from Tocris. To assess the local drug effects, compounds were dissolved in artificial cerebrospinal fluid (in mM: NaCl, 125; KCl, 2.5; CaCl_2_, 1.26 and MgCl_2_, 1.18) and administered by reverse dialysis at the stated concentrations (uncorrected for membrane recovery).

### 4.8. Behavioral Testing

Behavioral analyses were performed in non-Tg and A30P*A53T*α-Syn mice with intervals of 1–7 days between tests. Different behavioral paradigms were used to evaluate motor and cognitive functions as well as anxiety- and depressive phenotype. All tests were performed between 10:00 and 15:00 h by an experimenter who was blinded to mouse treatments. On the test day, mice were placed in a dimly illuminated behavioral room and were left undisturbed for at least 1 h before testing [33,36].

Open field test. Motor activity was measured in four Plexiglas open field boxes (35 × 35 × 40 cm) that were indirectly illuminated (25–40 lux) to avoid reflection and shadows. The floor of the open field boxes was covered with an interchangeable opaque plastic base that was replaced for each animal. Motor activity was recorded for 15 min by a camera connected to a computer (Video-track, Viewpoint, Lyon, France) [62]. The following variables were measured: horizontal locomotor and exploratory activity, defined as the total distance moved in cm including fast/large (speed > 10.5 cm/s) and slow/short movements (speed 3–10.5 cm/s), as well as the activation time including the mean speed (cm/s) and resting time (s), and the number of rearings.

Cylinder test. Mice were tested for motor asymmetry in the cylinder test. Each mouse was placed in an acrylic cylinder (diameter, 15 cm; height, 27 cm), and the total number of left and right forepaw touches performed, as well as the number of rearings in 5 min was counted. Behavioural equipment was cleaned with water after each test session to avoid olfactory cues.

Dark-light box test. The apparatus consisted of two glass boxes (27 × 21 cm) with an interconnecting grey plastic tunnel (7 × 10 cm). One of these boxes was painted in black and was weakly lit with a red 25-W bulb (42 lux). The other box was lit with a 60-W desk lamp (400 lux) placed 30 cm above the box, which provided the unique laboratory illumination condition. At the beginning of the test, mice were placed individually in the middle of the dark area facing away from the opening, and were videotaped for 5 min. The following variables were recorded: (a) time spent in the lit box, (b) the latency of the initial movement from the dark to the lit box, and (c) the number of entries in the lit box. A mouse was considered to enter the new area when all four legs were in this area. The floor of each box was cleaned between the mice.

Tail suspension test. Mice were suspended 30 cm above the floor by adhesive tape placed approximately 1 cm from the tip of the tail. Sessions were videotaped for 6 min and the immobility time was measured (Smart, Panlab, Cornellà, Spain).

Novelty object recognition test. Mice were placed into an open field box of Plexiglas (40 × 35 × 16 cm) with a soured of illumination (60 lux) in the center of the box. Two objects of identical shape were used as familiar objects. During the habituation period, the mice could freely explore the open field box without objects for 10 min on two consecutive days. In the acquisition test, the two identical objects were placed separately in the center of the open field and the mice explored it for 10 min. To minimize the presence of olfactory traces, the objects and the open field were cleaned with water between each trial. Twenty-four hours after the acquisition test, one of the familiar objects was replaced with a new object, and the amount of time spend exploring both objects (familiar and novel) was recorded for a period of 10 min. Exploration of an object was defined as pointing the nose at an object at a distance of <1 cm and/or touching it.

### 4.9. Statistical Analysis

All values are expressed as the mean ± standard error of the mean (SEM). Statistical comparisons were performed using GraphPad Prism 8.01 (GraphPad software, Inc., San Diego, CA, USA) using the appropriate statistical tests, as indicated in each figure legend. Outlier values were identified by the Grubbs’ test (i.e., Extreme Student zed Deviate, ESD, method) using GraphPad Prism software and excluded from the analysis when applicable. Differences among means were analyzed by either 1- or 2-way analysis of variance (ANOVA) or the two-tailed Student’s *t*-test, as appropriate. When ANOVA showed significant differences, pairwise comparisons between means were subjected to Tukey’s post hoc test or Sidak’s multiple comparisons test, as appropriate. Differences were considered significant at *p* < 0.05.

## Figures and Tables

**Figure 1 ijms-22-02939-f001:**
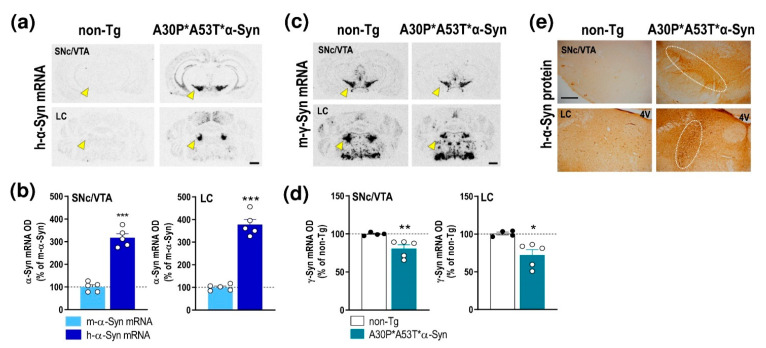
Profile of α-Syn expression in monoamine brain areas of A30P*A53T*α-Syn transgenic mice. (**a**) Coronal brain sections showing h-α-Syn mRNA levels in SNc/VTA assessed by in situ hybridization. Yellow arrowheads indicate the brain regions quantified in b. Scale bar: 1 mm. (**b**) Increased h-α-Syn mRNA expression in SNc/VTA and LC of A30P*A53T*α-Syn transgenic mice compared with m-α-Syn mRNA expression levels in the same mice (*n* = 5 mice/group; *** *p* < 0.001 vs. m-α-Syn mRNA levels; Student’s *t*-test). (**c**) Coronal brain sections showing m-γ-Syn mRNA levels in SNc/VTA and LC assessed by in situ hybridization. Yellow arrowheads show the lower density of m-γ-Syn expression in A30P*A53T*α-Syn mice compared with non-Tg mice. Scale bar: 1 mm. (**d**) Reduced m-γ-Syn mRNA expression in SNc/VTA and LC of A30P*A53T*α-Syn transgenic mice compared with non-Tg mice (*n* = 4–5 mice/group; * *p* < 0.05, ** *p* < 0.01 versus non-Tg mice; Student’s *t*-test). (**e**) Representative photomicrographs showing accumulated levels of h-α-Syn protein in SNc/VTA and LC of A30P*A53T*α-Syn transgenic mice. White frames indicate the SNc/VTA and LC brain regions. Scale bar: 200 µm. For all figures, data represent the mean ± SEM. Abbreviations: SNc, substantia nigra compacta; VTA, ventral tegmental area; LC, locus coeruleus; 4V, 4-ventricle. See also Appendix A.

**Figure 2 ijms-22-02939-f002:**
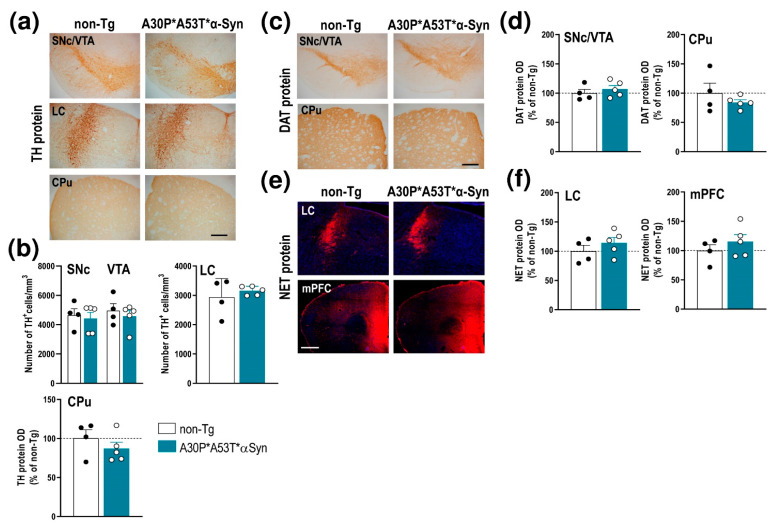
Integrity of DA and NE systems of middle-aged (5 months) A30P*A53T*α-Syn transgenic mice. (**a**) TH-immunostained brain sections containing SNc/VTA, LC and CPu. Scale bar: 200 µm. (**b**) No differences in the number of TH^+^ neurons were found in the SNc, VTA or LC of A30P*A53T*α-Syn vs. non-Tg mice. Likewise, the density of striatal TH^+^ terminals was comparable between both phenotypes. (**c**) DAT-immunostained brain sections containing SNc/VTA and CPu. Scale bar: 200 µm. (**d**) No differences in DAT protein density were found in the SNc, VTA or CPu of A30P*A53T*α-Syn vs. non-Tg mice. (**e**) Confocal images showing the NET protein density in LC and mPFC of A30P*A53T*α-Syn and non-Tg mice. Scale bar: 200 µm. (**f**) No differences in NET protein density were found in LC and mPFC of both phenotypes. Data are represented as mean ± SEM, *n* = 4–5 mice/group. Abbreviations: mPFC, medial prefrontal cortex; CPu, caudate putamen; HPC, hippocampus; SNc, substantia nigra compacta; VTA, ventral tegmental area; DR, dorsal raphe nucleus; LC, locus coeruleus.

**Figure 3 ijms-22-02939-f003:**
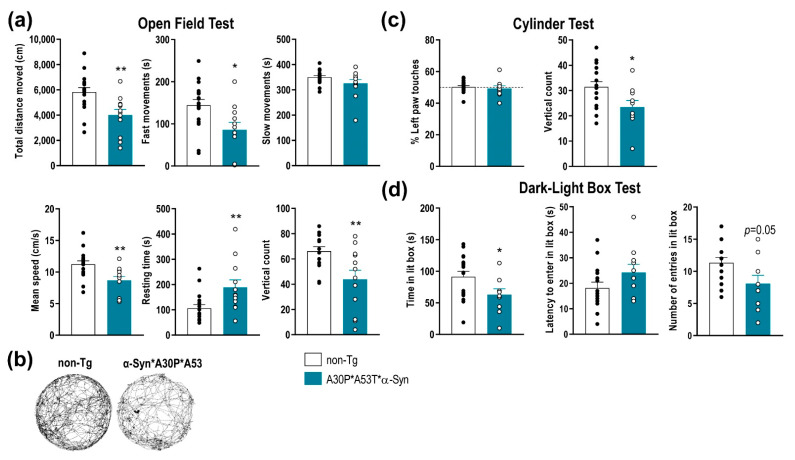
Impairment of motor function and anxiety-like behaviors in A30P*A53T*α-Syn transgenic mice. (**a**) Comparison of spontaneous locomotor activity between 5-month-old A30P*A53T*α-Syn and non-Tg mice in the open field test. Total distance traveled, fast movements, average speed and vertical count were significantly decreased in A30P*A53T*α-Syn mice compared with non-Tg mice. In parallel, transgenic mice showed an increased resting time compared with non-Tg mice. (**b**) Representative locomotor activity tracking was obtained from both phenotypes. (**c**) No difference in motor asymmetry examined by the use of the left paw was detected. However, A30P*A53T*α-Syn mice showed a reduced vertical count in the cylinder test compared with non-Tg mice. (**d**) In the dark-light box test, A30P*A53T*α-Syn transgenic mice evoked an anxiety-like response compared with non-Tg mice, as shown by the shorter spent time in the lit box but the marginal effects in the latency period to entering in the light box or the number of entries. Data are the mean ± SEM, *n* = 10–15 mice/group. Student’s *t-*test, * *p* < 0.05, ** *p* < 0.01 compared with non-Tg mice. See also Appendix A.

**Figure 4 ijms-22-02939-f004:**
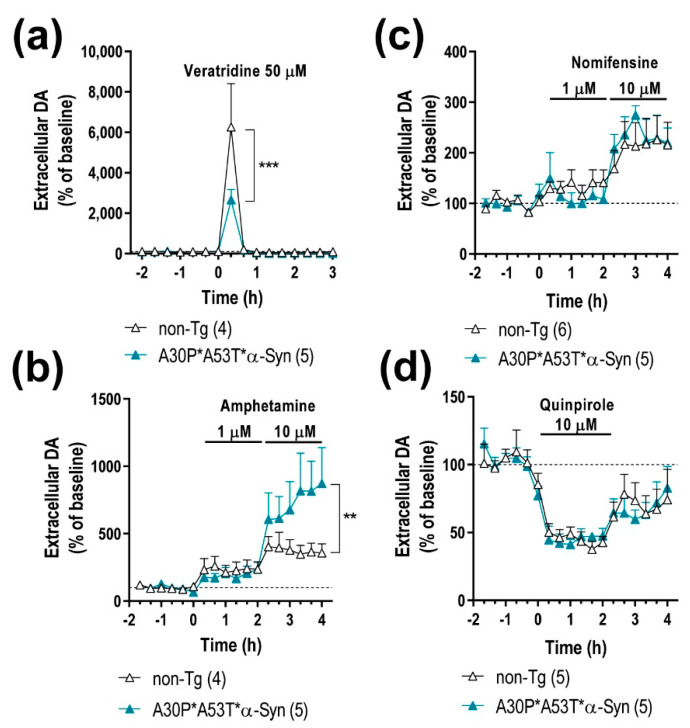
A30P*A53T*α-Syn overexpression in TH^+^ neurons alters DA neurotransmission in the nigrostriatal pathway. (**a**) Local veratridine infusion (depolarizing agent; 50 μM) significantly increased DA release in the caudate putamen (CPu) of both mouse phenotypes. However, this effect was significant smaller in A30P*A53T*α-Syn mice than in non-Tg mice. (**b**) Direct application of amphetamine (DA releaser and DAT inhibitor; 1 and 10 μM) by reverse dialysis induced increases in DA release in the CPu, with the effect being significantly more pronounced in A30P*A53T*α-Syn mice than in non-Tg mice. (**c**) However, local nomifensine infusion (DAT inhibitor; 1 and 10 μM) dose-dependently increased the extracellular DA concentration in CPu, with this effect comparable being in both phenotypes. (**d**) Local activation of DA D2 receptors with quinpirole (DA D2 agonist, 10 μM) similarly decreased striatal DA release in A30P*A53T*α-Syn and non-Tg mice. Data are expressed as the mean ± SEM, *n* = 4–6 mice/group as indicated in the parenthesis. Two-way ANOVA and Tukey’s multiple comparisons test, ** *p* < 0.01, *** *p* < 0.001 compared with non-Tg mice.

**Figure 5 ijms-22-02939-f005:**
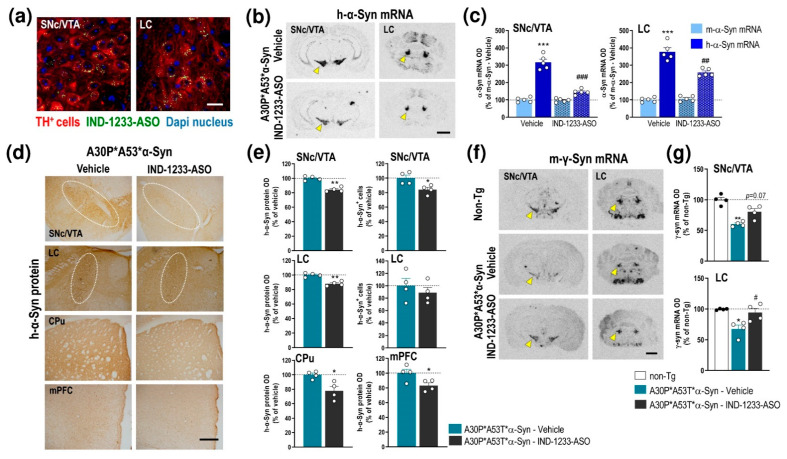
Intracerebroventricular IND-1233-ASO treatment selectively inhibits the expression and accumulation of h-α-Syn in interconnected DA and NE brain regions of A30P*A53T*α-Syn transgenic mice. (**a**) Confocal images showing the co-localization of A488-IND-1233-ASO (green) with TH^+^ neurons (red) in the SNc/VTA and LC. Cell nuclei were stained with DAPI (blue). Scale bar: 10 µm. (**b**) Coronal brain sections showing reduced h-α-Syn mRNA expression in the SNc/VTA and LC of A30P*A53T*α-Syn mice treated with IND-1233-ASO (100 µg/day for 28 days), as assessed by in situ hybridization. Yellow arrowheads show h-α-Syn mRNA expression in SNc/VTA and LC. Scale bar: 1 mm. (**c**) Bar graphs showing significant reductions in h-α-Syn mRNA levels, but not m-α-Syn mRNA, in SNc/VTA and LC of IND-1233-ASO-treated A30P*A53T*α-Syn mice (*n* = 5 mice/group; *** *p* < 0.001 vs. m-α-Syn mRNA levels; ^##^
*p* < 0.01, ^###^
*p* < 0.001 vs. vehicle-treated A30P*A53T*α-Syn mice; two-way ANOVA and Tukey’s multiple comparisons test). (**d**) Representative photomicrographs showing a lower h-α-Syn protein density in the SNc/VTA, LC, CPu and mPFC of IND-1233-ASO-treated A30P*A53T*α-Syn mice vs. vehicle-treated A30P*A53T*α-Syn mice. White frames indicate SNc/VTA and LC brain regions. Scale bar: 200 µm. (**e**) Bar graphs showing significant reductions in h-α-Syn protein levels in the brain areas analyzed, as well as a decreased number of h-α-Syn^+^ cells in the SNc/VTA of IND-1233-ASO-treated A30P*A53T*α-Syn mice (*n* = 4 mice/group; * *p* < 0.05, ** *p* < 0.01 versus vehicle-treated A30P*A53T*α-Syn mice; Student’s *t*-test). (**f**) Representative images showing increased m-γ-Syn mRNA expression in SNc/VTA and LC of A30P*A53T*α-Syn mice treated with IND-1233-ASO as assessed by in situ hybridization. Yellow arrowheads show m-γ-Syn mRNA expression in SNc/VTA and LC. Scale bar: 1 mm. (**g**) Bar graphs showing significant increases in m-γ-Syn expression in the LC and a marginal effect in the SNc/VTA of IND-1233-ASO-treated A30P*A53T*α-Syn mice (*n* = 4 mice/group; * *p* < 0.05, ** *p* < 0.01 vs. non-Tg mice; ^#^
*p* < 0.05 vs. vehicle-treated A30P*A53T*α-Syn mice; one-way ANOVA). For all figures, data are the mean ± SEM. Abbreviations: mPFC, medial prefrontal cortex; CPu, caudate putamen; SNc, substantia nigra compacta; VTA, ventral tegmental area; LC, locus coeruleus. See also Appendix A.

**Figure 6 ijms-22-02939-f006:**
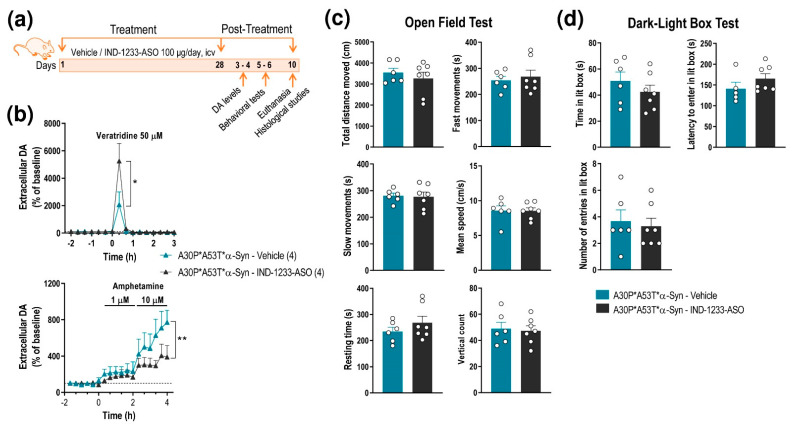
Intracerebroventricular IND-1233-ASO treatment recovers DA neurotransmission deficits, but not behavioral phenotype, in A30P*A53T*α-Syn transgenic mice. (**a**) Treatment timeline. A30P*A53T*α-Syn transgenic mice (4 months of age) received randomly vehicle or IND-1233-ASO (100 µg/day) into the lateral ventricle for 28 days using osmotic minipumps. In a group of mice, DA release was examined using intracerebral microdialysis at 3–4 days post-treatment, while behavioral tests were performed in an additional group at 5–6 days post-treatment. All mice were sacrificed on day 10 post-treatment. (**b**) Microdialysis approach using veratridine and amphetamine agents, as shown in Figure 3, confirmed the normalization of DA neurotransmission in the CPu of A30P*A53T*α-Syn mice treated with IND-1337-ASO (100 μg/day for 28 days) compared with transgenic mice treated with a vehicle. (**c**,**d**) No significant differences were detected in the open field test (**c**) or the dark-light box test (**d**) between both groups treated with IND-1337-ASO or a vehicle. Data are expressed as the mean ± SEM. Number of mice used in each procedure is indicated in parenthesis. Two-way ANOVA and Tukey’s multiple comparisons test, * *p* < 0.05, ** *p* < 0.01 compared to A30P*A53T*α-Syn mice treated with a vehicle.

**Table 1 ijms-22-02939-t001:** Baseline DA and DOPAC dialysate concentrations in the CPu of mice.

Mice	Experimental Conditions	Baseline DA	Baseline DOPAC
non-Tg	aCSF	10.3 ± 1.6 (15)	0.8 ± 0.1 (15)
	aCSF + DMSO 1%	7.2 ± 2.2 (4)	1.6 ± 0.4 (4)
A30P*A53T*α-Syn	aCSF	8.5 ± 1.2 (15)	1.1 ± 0.1 (15)
	aCSF + DMSO 1%	6.6 ± 1.7 (5)	1.8 ± 0.5 (5)

Extracellular DA and DOPAC levels are expressed as the fmol/20-min fraction. In the experiments involving the evaluation of the veratridine effect on extracellular DA and DOPAC levels, dimethyl-sulfoxide (DMSO) was added to the artificial cerebrospinal fluid (aCSF), respectively. Data are the mean ± SEM of the number of mice shown in parentheses.

## Data Availability

The data presented in this study area available on request from the corresponding author. The data are not publicly available due to privacy.

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
