# Peer review of "Intracerebral Administration of a Ligand-ASO Conjugate Selectively Reduces α-Synuclein Accumulation in Monoamine Neurons of Double Mutant Human A30P*A53T*α-Synuclein Transgenic Mice"

_ijms, 2021, doi:10.3390/ijms22062939_

Round 1

Reviewer 1 Report

The authors tested antisense oligonucleotide able to block expression of Parkinson's disease related protein α-synuclein in transgenic mice producing human α-synuclein with two disease-related mutations. The oligonucleotide was conjugated with indatraline producing a compound (IND-1233-ASO) that is selectively delivered to monoamine neurons (including ones crucial for Parkinson's disease development). The key compound, IND-1233-ASO, was already validated by the authors in previous works. The used A30P*A53T*α-Syn transgenic mice line was also already established.

The transgenic mice clearly show a decreased locomotor activity and less pronounced anxiety-like behavior. Unfortunately, treatment with IND-1233-ASO was insufficient to significantly reverse the behavioral impairments observed in the transgenic mice.

This work is performed on a very good level and presents careful description of experimental results and techniques.

I think it should be published after slight improvement of the presentation:

1) Conclusions should be toned down. Namely I do not see data confirming that "Reduction of α-Syn synthesis would improve deficits in DA neurotransmission in PD by preferentially enhancing and normalizing a deficient DA release. Overall, these effects would potentially delay the progression of PD symptomatology without serious deleterious effects."

2) This manuscript describes a lot of experimental results obtained by a quite large team and I think, it is necessary to write "Author Contribution" section more precisely, namely specify people responsible for particular experiments.

3) Figures looks too crowded, some images are too small, and captions are overloaded with data.  It would be nice to improve their quality and may be move non-essential parts to SI in order to focus reader's attention at the most important results.

Minor points

It would be nice to show the whole structure of IND-1233-ASO in the paper or SI. It is not clear how stable is the bond between indatraline and oligonucleotide part.

Figure 3 is quite messy and its caption is too long and partially duplicates the text. I think it would be more appropriate to present the data as a table (for example columns Parameter|nonTg|A30P*A53T*αSyn|significance/comments,   and separate by subsections for each test)

Abstract. Statement "Point mutations and multiplications of the α-Syn encoding SNCA gene correlate with early-onset PD, thus leading to a solid premise for developing therapies that inhibit α-Syn synthesis" is too strong

L44         Ref [2] is not appropriate to show how elderly population is affected by PD.

L63         Is ref[1] appropriate here? It looks like it does not deal with mice model.

L102       typo in section title

L119      Some text in Fig 1 is hard to read.

                May be in Fig 1a it would be better to show only half of sections (left=nonTg, right= A30P*A53T*αSyn) to economize the space. Fig 1c can be moved to SI.

L178      (Fig 3) check text in panel (b)

L273       Fig 5a green color is not visible. Please provide a better proof of co-localization (red-green distribution plot, for example). Fig 5e text is too small.

L598      It would be more appropriate to write the used excitation wavelengths rather than list all lasers present in microscope

L646      Is "red 25-W bulb (0 lux)" correct?

L656      Rephrase the sentence

Reviewer 2 Report

The authors made an interesting finding that α-Synulcein-targeted oligonucleotides conjugated with indatralin- (IND-1233-ASO), when administered intracerebrally, can reduce the accumulation in SNCA gene. They used a modern methodology to confirm this fact and obtained reliable data. It can be stated that, the manuscript is prepared reliably, interesting results have been obtained, which are fairly interpreted. I have no objections to the experimental part of the work and the interpretation of the results.

Already when I took on the role of reviewer, I indicated that I am a well-informed layman in this area. I can assess the quality of experimental work and its interpretation. I cannot assess the role of α-synulcein in the etiopathogenesis of PA in the light of current knowledge. Therefore I have not specific comments.

However, I consider the scientific merit of the contribution to be very significant.

Reviewer 3 Report

A brief summary

The authors used a compound (an antisense oligonucleotide) that selectively reduces α-syn synthesis in midbrain dopamine and norepinephrine neurons. They used a double mutant transgenic mouse model, specifically the human mutant A30P and A53T point mutations in α-syn, to test whether their compound (IND-1223-ASO) could downregulate human α-syn levels. Using chronic infusion to the brains of the transgenic mice, they successfully reduced human α-syn levels in Parkinson’s disease-relevant brain areas. Further, the authors showed normalization of dopamine neurotransmission with the treatment.

Broad comments

  • Authors rightly emphasize that this treatment did not result in a severe reduction in α-syn levels in the model and how this is important for a therapy going forward. However, one thing that is unclear is when the behavioural tests and other outcome measures were performed after treatment. This needs to be stated clearly in results and/or methods as it is important for whether this was an acute effect of the treatment or whether the effects lasted after treatment cessation.
  • A timeline of how the experiments were performed, indicating start and duration of treatment, when behavioural tests were performed, and the end of the experiment, would be useful to include here. Additionally, the age of the mice at start and end of treatment should be mentioned somewhere.
  • Could the authors comment on whether this treatment would need to be administered i.c.v. throughout a patient’s lifespan (if indeed the results shown are from immediately after treatment cessation)?
  • It is good that the authors note there is no acute toxicity, and no cell loss is observed with the treatment, but have longer studies been performed (with or without continued treatment) to see later effects? Was weight measured in these mice throughout and after treatment? That should be included here or referenced from elsewhere.
  • Authors should comment as to why i.c.v. administration was used here when they have previous positive results with intranasal administration. Especially since intranasal administration is more clinically relevant.
  • It should be mentioned why 5-month-old animals were used (as opposed to older), where it has previously been observed that older animals show a more overt phenotype and Parkinson’s disease is a disease of ageing (also, treatments are often administered to patients later in disease stage). What is the advantage of using 5-month-old mice here, authors should justify how it answers their research question.
  • Authors refer to motor “deficits” in the transgenic mice, however the phenotype observed is not very robust, but interesting nonetheless. They need to put a caveat here as the results (less total difference travelled, more resting time, and where the mice were travelling) could also relate to the anxiety-like phenotype. Although the authors show a difference in “fast movements” and mean speed as well, but it’s unclear that this relates directly to a motor deficit per se. Could the authors expand on this? The paw touches from cylinder test are shown, and unsurprisingly there was no difference in motor asymmetry, however the authors should show total number of rearings. Were other motor tests performed? It is difficult to conclude from this that there was a clear motor deficit, but rather a change in motor behaviour along with the anxiety-like phenotype. Along these lines, the mention of “motor deficits” in the abstract should be modified.
  • The authors show thorough studies with microdialysis which expands on the current knowledge of how α-syn may be affecting neurotransmission. Additionally, were overall tissue dopamine levels checked in any brain areas, particularly in vehicle vs. treatment? This would be potentially interesting information to add to the paper.
  • These α-syn transgenic models can have varied outcomes depending on the genetic background, as authors somewhat allude to in discussion, could they emphasize briefly in the discussion when they are mentioning other studies using this model if it differs in this regard as this is important for the translational value of the research going forward. Some discussion about how the model used is useful when testing this type of drug for human sporadic Parkinson’s disease or other synucleinopathies should also be included.
  • Why did authors only use male mice? Female mice should also be used or a justification as to why not.
  • The English is good and does not take away from the understanding of the paper, however it would be good to have someone read it through to correct various wording mistakes that obscure some sentences, particularly in the discussion.

Specific comments 

  • Authors refer to 5-month-old mice as “middle aged”, would rather use “adult” here as these mice live longer than 10 months.
  • Please also mention age and sex of mice in the abstract.
  • Rather than having solid bars on graphs only, authors should indicate each mouse with its own data point throughout the results.
  • Figure 1f: How many mice were used to get these representative images, was this result consistent across animals?
  • Figure 5a: It is not possible to see the DAPI and the green channel is very faint. Authors should make the panels larger or otherwise use different colours.
  • Figure 5d: Panels could be larger here so it is more clear.
  • For Figures 5 and 6, are these the same mice? There are a few more animals in the behavioural tests than the other outcomes. Also, there are about half as many as in Fig. 3 for behaviour. Did you have a transgenic mouse group with no infusion/sham operation for comparison to the vehicle and treatment groups that was measured at the same time? This should be clarified in the results section.
  • Figure 6: Why are there 4 mice/group for the microdialysis experiments and 5-7/group in the behavioural tests?
  • The supplementary figure indicating no loss of cells could be put to the main manuscript.
  • Methods 4.1: How many animals were used in total needs to be mentioned.
  • Methods 4.3: Authors do not mention painkillers or post-operative care at all, this absolutely needs to be specified. Also, how the mice were assigned to treatment groups.
  • Line 520: What was the vehicle used?
  • Blinding is mentioned for the behavioural analysis, please also mention this in the other sections where analysis was performed, assuming it was done there as well.
  • Line 578: I assume that both sides of the brains were counted, this could be mentioned.
  • Line 635: 15 minutes is a less common measurement time on open field, could the authors put a reference or state why this was done (as opposed to 60 mins or more)?

Round 2

Reviewer 3 Report

I am happy with the changes made to the manuscript and would accept it now.